# Evolution of Angiogenic Factors in Pregnant Patients with Breast Cancer Treated with Chemotherapy

**DOI:** 10.3390/cancers13040923

**Published:** 2021-02-23

**Authors:** Cristina Saura, Olga Sánchez, Sandra Martínez, Carmen Domínguez, Rodrigo Dienstmann, Fiorella Ruíz-Pace, Maria Concepció Céspedes, Ángeles Peñuelas, Javier Cortés, Elisa Llurba, Octavi Córdoba

**Affiliations:** 1Medical Oncology Department, Vall d’Hebron University Hospital, Vall d’Hebrón Institute of Oncology (VHIO), 08035 Barcelona, Spain; csaura@vhio.net (C.S.); apenuelas@vhio.net (Á.P.); 2Maternal and Child Health and Development Network (SAMID), RD12/0026/16 and RD16/0022/0015, Institute of Health Carlos III, 28029 Madrid, Spain; osanchezg@santpau.cat; 3Biochemistry and Molecular Biology Research Centre for Nanomedicine, Vall d’Hebron Research Institute, 08035 Barcelona, Spain; mcdluengo@gmail.com; 4Vall d’Hebrón Institute of Oncology (VHIO), 08035 Barcelona, Spain; sandra.martinez@ftmoreto.org; 5Center for Biomedical Research on Rare Diseases (CIBERER), 08028 Barcelona, Spain; 6Oncology Data Science (ODysSey), Vall d’Hebrón Institute of Oncology (VHIO), 08028 Barcelona, Spain; rdienstmann@vhio.net (R.D.); fruiz@vhio.net (F.R.-P.); 7Pediatrics Department, Vall d’Hebron University Hospital, 08035 Barcelona, Spain; mcespede@vhebron.net; 8Vall d’Hebrón Institute of Oncology (VHIO), International Breast Cancer Center, Quiron Group, 08035 Barcelona, Spain; jacortes@vhio.net; 9Department of Obstetrics, Maternal-Fetal Medicine Unit, Santa Creu i Sant Pau University Hospital, Universitat Autònoma de Barcelona, 08025 Barcelona, Spain; 10Obstetrics and Gynecology Department, Hospital Universitari Son Espases, 07120 Palma, Spain; octavi.cordoba@ssib.es; 11Institut d’Investigació Sanitària de les Illes Balears (IdISBa), 07120 Palma, Spain

**Keywords:** breast cancer, pregnancy, angiogenic factors, chemotherapy

## Abstract

**Simple Summary:**

Anthracyclines and taxanes are being used as a standard treatment for breast cancer diagnosed during pregnancy. These chemotherapy regimens allow the continuation of pregnancy without delaying cancer treatment with relatively good maternal and neonatal outcomes. However, their effects on placental function and fetal development are not completely understood. Maternal serum angiogenic factors are a surrogate of placental function and are abnormal weeks before placental complications such as preeclampsia or intrauterine growth restriction development. In our cohort, pregnant women with breast cancer treated with chemotherapy during pregnancy show an antiangiogenic state with significantly higher levels of soluble fms-like tyrosine kinase (sFlt-1), sFlt-1/PGF ratio, and soluble endoglin (sEng) at the end of the third trimester. Angiogenic factors could be useful in the clinical obstetric management of these patients, although more studies are guaranteed.

**Abstract:**

High prevalence of placental-derived complications, such as preeclampsia and intrauterine growth restriction, has been reported in women with breast cancer (BC) treated with chemotherapy during pregnancy (PBC-CHT). Aim: To ascertain whether PBC-CHT is associated with an imbalance of angiogenic factors, surrogate markers for placental insufficiency, that could explain perinatal outcomes. Methods: Prospective study between 2012 and 2016 in a single institution. Soluble fms-like tyrosine kinase (sFlt-1), placental growth factor (PlGF), and soluble endoglin (sEng) in maternal blood were assessed throughout pregnancy in 12 women with BC and 215 controls. Results: Cancer patients were treated with doxorubicin-based regimes and with taxanes. Ten PBC-CHT (83%) developed obstetrical complications. At the end of the third trimester, significantly higher levels of sFlt-1; sFlt-1/PGF ratio, and sEng levels were observed in BC women as compared to controls. Moreover; there was a significant correlation between plasma levels of sFlt-1 and the number of chemotherapy cycles administered. Besides, more chemotherapy cycles correlated with lower birthweight and head circumference at birth. Conclusions: Women with BC treated during pregnancy showed an antiangiogenic state compatible with placental insufficiency. Angiogenic factors could be useful in the clinical obstetric management of these patients; although further studies will be required to guide clinical decision-making.

## 1. Introduction

Breast cancer diagnosed during pregnancy represents a complex set of challenges for the patient and the clinicians, as maternal benefit should be balanced with fetal risk. Cancer diagnosed during pregnancy is a rare occurrence with an estimated frequency of 2.3 cases per 100,000 deliveries [1]. However, it is increasing due to the rise in maternal age [2] and breast cancer incidence worldwide [3].

Use of anthracyclines and taxanes is safe during second and third trimester of pregnancy [4,5,6,7,8,9,10,11,12]. However, some obstetric complications, so far not related to antineoplastic treatment, such as preeclampsia, small for gestational age, and spontaneous preterm delivery, have been described [10,13] and there is still limited information regarding cytotoxic effects on placenta function and fetal wellbeing.

According to animal models, placenta has a barrier function against chemotherapy. Levels of doxorubicin in baboon fetal plasma are 7.5 % (+/− 3.2%) of the maternal levels [14]. Despite this barrier function, the placenta may be injured. We hypothesized that this potential injury by chemotherapy may be responsible for the obstetrical complications described in cancer patients treated during pregnancy.

The dysregulation of angiogenesis in the placenta and maternal–fetal circulation has emerged as one of the main pathophysiological features in the development of placental insufficiency and its clinical consequences. In women developing preeclampsia (PE) or/and intrauterine growth restriction (IUGR), angiogenic factor placental growth factor (PlGF), and anti-angiogenic factor soluble fms-like tyrosine kinase (sFlt-1), and its relation, are a clinically useful tool for the prediction, prognosis, and management of these complications [15,16]. Angiogenic factors as surrogate markers of placental insufficiency may help to assess placental damage in pregnant women treated with chemotherapy during pregnancy (PBC-CHT) and potentially guide management and delivery planning.

The aim of the present study is to ascertain whether pregnant women with breast cancer treated with chemotherapy have an imbalance of angiogenic factors towards a pro-antiangiogenic state that could be related to perinatal outcomes.

## 2. Patients and Methods

### 2.1. Study Design

All consecutive patients with breast cancer diagnosed during pregnancy and treated at Vall d’Hebron University Hospital with CHT between October 2012 and June 2016 were eligible to participate in a prospective case-control study. The eligibility criteria for study entry were histological diagnosis of primary or recurrent breast cancer and starting of treatment with CHT during pregnancy (second and third trimester). All subjects gave their informed consent for inclusion before they participated in the study. This study was conducted in accordance with the Declaration of Helsinki, and the protocol was approved by the Ethics Committee of the Vall d’Hebron University Hospital (PR(AMI) 83/2012). Treatment strategy was decided case by case in a multidisciplinary team evaluation and according to standardized international recommendations [11,12,13]. Twin pregnancies were excluded. sFlt-1, PlGF, and sEng in maternal blood were assessed at diagnosis of breast cancer, and each time before and after CHT treatment throughout pregnancy and at delivery. Longitudinal changes in angiogenic factors in affected pregnancies were compared with those of low-risk pregnancies at the same gestational age (ratio 1 case:18 controls).

### 2.2. Study Procedures

Pregnant women with breast cancer were followed in the Breast cancer pregnancy unit that was composed by a Maternal–Fetal specialist, an Oncologist, and a Gynecologic surgeon. After the diagnosis of breast cancer and before and after CHT treatment, women have an exhaustive obstetrical and fetal evaluation that includes: blood pressure, weight, maternal wellbeing, and obstetric abdominal and vaginal ultrasound study using 6–4 MHz probes (Siemens Sonoline Antares, Siemens Medical, Erlangen, Germany). Estimated fetal weight, amniotic fluid index, and Doppler velocimetry; uterine artery pulsatility index (PI), umbilical artery PI and median cerebral artery PI, and maximum peak systolic velocity (Vmax MCA) were evaluated. Following obstetric evaluation, medical oncologist assesses whether potential toxicities derived from chemotherapy existed. Subsequently, if both evaluations (obstetrical and oncological) were correct, maternal blood was obtained and chemotherapy cycle was administered. The frequency of the blood tests was established by the chemotherapy schedule defined appropriately for each patient based on her tumor stage, patient’s characteristics, and weeks of pregnancy at the time of diagnosis. If there were concerns regarding maternal or fetal health, termination or delay of the chemotherapy were considered by the multidisciplinary team, or delivery was planned. If no specific concerns during treatment, delivery was planned and induced according to gestational age and CHT schedule to minimize any delay in maternal treatment. At the time of delivery, maternal and fetal cord blood was also obtained. Clinicians were blinded to angiogenic factors results.

Outcomes: Diagnostic criteria for preeclampsia were new onset of both hypertension (systolic blood pressure of 140 mmHg or above and/or diastolic blood pressure of 90 mmHg or above) and proteinuria (protein dipstick urinalysis result of 2+ or greater, or 300 mg protein per 24-h urine collection or greater, or protein in spot urine greater than or equal to 30 mg/dL, or protein/creatinine ratio greater than or equal to 30 mg/mmol) after 20 weeks’ gestation [17]. Suspicion of fetal anemia was defined if fetal Vmax MCA >1.55 MoM according to Mari et al. [18]. Fetal adverse outcomes were perinatal/fetal death, delivery before 34 weeks, intrauterine growth restriction, placental abruption, respiratory distress syndrome, necrotizing enterocolitis, and intraventricular hemorrhage.

### 2.3. Samples

Maternal venous blood was drawn for routine blood tests and samples were processed within 1 h. Plasma was separated by centrifugation at 2000· *g* for 5 min at 4 °C, and sample aliquots were immediately stored at −80 °C until assayed.

### 2.4. Plasma Levels of Angiogenic and Anti-Angiogenic Factors

Soluble fms-like tyrosine kinase (sFlt-1), free placental growth factor (PlGF), and soluble endoglin (sEng) concentrations were measured under pregnancy type-blinded conditions. Enzyme-linked immunosorbent assays (ELISA) for human sFlt-1, PlGF, and sEng were performed in duplicate using commercial kits (R&D Systems Europe Ltd., Abington, UK) following the manufacturer’s instructions. Minimum detectable values in the assays were 3.5 pg/mL for sFlt-1, 7 pg/mL for PlGF, and 0.007 ng/mL for sEng. In all kits, intra-assay precision was always < 5% and inter-assay precision < 10%. Linear regression coefficients of the standard curves were never < 0.99%.

### 2.5. Statistical Analysis

Statistical analyses were performed using Prism software (GraphPad, version 5.02, San Diego, CA, USA). The distribution of continuous variables was tested for normality by the D’Agostino and Pearson omnibus K2 test. Demographic and clinical data were compared by Fisher’s exact test for categorical variables and independent-sample Student’s *t*-test or Mann–Whitney U-test, as appropriate, for continuous variables.

Differences in the evolution of angiogenic factors between groups were analyzed by two-way analysis of variance and Bonferroni post-test.

Pearson’s or Spearman’s correlation analysis, as appropriate, was undertaken to relate some variables in pregnant women affected by breast cancer.

The *p*-value threshold used for significant differences was <0.05.

## 3. Results

### 3.1. Patient Characteristics

Between October 2012 and June 2016, 12 women with breast cancer diagnosed during pregnancy were treated with chemotherapy at Vall d’Hebron University Hospital. A total of 215 women were included as the control population. Those women had normal pregnancies with no risk factors and were recruited in Vall d’Hebron obstetrics clinics. Table 1 summarizes demographic characteristics of the study population. Maternal age was significantly higher in pregnant women affected by breast cancer (PBC-CHT) than in healthy pregnant women (P-CTRL) (*p* = 0.0012).

Values expressed as Mean ± SD or number (percentage). Comparisons between pregnant women with breast cancer and treated with chemotherapy (PBC-CHT) and the control group (P-CTRL) were analyzed by Fisher’s exact test for categorical variables and independent-sample Student’s *t*-test or Mann–Whitney U-test, as appropriate, for continuous variables. BMI, body mass index.

The baseline characteristics of patients with breast cancer are listed in a table (Appendix A). A lump was detected through breast self-examination in all patients. Breast ultrasound was performed in all patients and suspicious alterations were found in all of them. Mammography was also performed in all patients. One patient was classified as non-informative (BIRADS 0), two patients (17%) were classified into BI-RADS 1 category (negative findings), four patients (33%) in category 4–5 (suspicious/suggestive of malignancy findings), and five patients (42%) in category 6 (biopsy with proven malignancy before conducting the mammography). The mean gestational age at diagnostic was 16 weeks (range 4–29). Five patients were diagnosed in the first trimester (42%), five in the second trimester (42%), and 2 in the third trimester (16%). One of them had metastatic breast cancer (stage IV). Forty-two percent of patients had hormone receptor-positive tumors (estrogen -ER- and/or progesterone receptor -PR- positivity of >1%), 42% overexpressed HER-2 (+3 result by immunohistochemistry or +2 by immunohistochemistry and in situ hybridation ISH/FISH positive) and 34% were classified as triple-negative breast cancer (ER, PR and HER-2 negative). Infiltrating ductal carcinoma was observed in most patients analyzed (92%).

### 3.2. Chemothepary Treatment

All patients were treated with doxorubicin-based regimes (AC, anthracycline plus cyclophosphamide) and 58% of them were also treated with taxanes. Eight patients began their chemotherapy during the second trimester of pregnancy and four patients in third trimester. Six patients received adjuvant treatment after surgery, five received neoadjuvant treatment, and one patient required chemotherapy for metastases. The patients received during pregnancy a median of 4 AC cycles (range 2–6) with a treatment length of 12 weeks (median, range 6–19) and a cumulative dose of doxorubicin of 217.5 mg/m^2^. Patients treated with taxanes received a median of 7 cycles of weekly paclitaxel (range 2–10) and a cumulative dose of 514.3 mg/m^2^. The mean time interval between the last cycle of chemotherapy and delivery was 2 weeks (range 0.7–5.71). Ten patients had some complications during treatment and during pregnancy. No patient presented unexpected or serious toxicity. Neutropenia grade G3 was reported in one patient and the decision was to delay the next cycle of chemotherapy. Detailed information is presented elsewhere (Appendix A).

### 3.3. Angiogenic Factors Levels in Cases and Control Women

A cross-sectional analysis of samples obtained at different gestational ages during pregnancy was made to evaluate gestational patterns of angiogenic and anti-angiogenic factors. At the end of the third trimester, sFlt-1 levels were significantly higher in pregnant women affected by breast cancer than in the control group (5792 ± 794 vs. 3801 ± 344 pg/mL, respectively), Figure 1. PlGF levels followed an inverse pattern at the end of the third trimester (334 ± 25 pg/mL in PBC-CHT vs. 613 ± 41 pg/mL in controls), Figure 2. Similarly, pregnant women with chemotherapy have an increase in sFlt1/PlGF ratio at the end of pregnancy (37.5 ± 11.3 in PBC-CHT vs. 10.3 ± 1.7 in controls), Figure 3. sEng levels were higher throughout gestation in women affected by breast cancer (Figure 4).

### 3.4. Perinatal Outcomes

Gestational age at delivery and birth weight were significantly lower in pregnant women affected by breast cancer (*p*-value = 0.0001 and *p*-value = 0.0001, respectively) (Table 2). Moreover, the PBC-CHT group presented a higher incidence of small for gestational age (SGA) babies (16%). Labor induction was recommended in 6 patients of PBC-CHT group to facilitate the continuation of chemotherapy treatment, the remaining patients of this group were induced for: (a) obstetrical complications (1 preterm labor and 1 preterm rupture of membranes), (b) maternal complications (disease progression in 2 patients), or (c) fetal complications (1 suspicion of fetal anemia and 1 patient with increased uterine and umbilical artery pulsatility index -PI).

Values expressed as Mean ± SD or number (percentage). Comparisons between pregnant women with breast cancer and treated with chemotherapy (PBC-CHT) and the pregnant women control group (P-CTRL) were analyzed by Fisher’s exact test for categorical variables and independent-sample Student’s t-test or Mann–Whitney U-test, as appropriate, for continuous variables. GA, gestational age; p, percentile; SGA, small for gestational age (Birth weight < 10th percentile).

Correlation analysis among maternal angiogenic factors at delivery and cycles of chemotherapy (n CHT), birth weight, or percentile was undertaken (Table 3). A significant correlation between plasma levels of sFlt-1 and the number of cycles administered was detected.

A Pearson’s correlation analysis was undertaken to relate number of cycles of chemotherapy administered (n CHT), birth weight, or percentile to maternal blood sFlt-1, PlGF, and sEng at delivery. A Spearman’s correlation analysis was undertaken to relate number of cycles of chemotherapy administered (n CHT), birth weight, or percentile to ratio sFlt-1 to PlGF at delivery.

A negative and significant correlation between the number of cycles of chemotherapy administered to pregnant women affected by breast cancer and birth weight of their children was observed (Figure 5). A negative correlation between the number of cycles of chemotherapy administered to pregnant women affected by breast cancer and neonate head circumference was also detected (Figure 6).

## 4. Discussion

To our knowledge, this is the first study that analyzes circulating levels of angiogenic factors in pregnant women with breast cancer. Our major findings are: (1) an angiogenic imbalance, towards antiangiogenic state, is developed at the third trimester of pregnancy and (2) birth weight at delivery inversely correlates with the cycles of chemotherapy administered. Treatment of breast cancer during pregnancy encompasses many therapeutic dilemmas. A multidisciplinary approach to the management of BC during pregnancy is critical to optimizing outcomes for both mother and fetus. It is widely accepted that chemotherapy administration should be avoided during the first trimester when organogenesis takes place [13,19].

Throughout the second and the third trimester of pregnancy, indications for neoadjuvant and adjuvant treatments are the same that recommended for non-pregnant women. Most experience with chemotherapy during pregnancy is from doxorubicin and taxane-based regimens. Indeed, the dosages are administered similar to non-pregnant women adjusting for weight gain without dose modifications with curative intent [20,21]. This exposure is associated with the risk of IUGR, prematurity, and low birth weight [7,22]. Loibl and colleges demonstrated that infants exposed to chemotherapy in utero had lower birth weight at the same gestational age than infants not exposed to chemotherapy [10]. The same trend was observed in our study, since birth weight, adjusted by gestational age, was significantly lower in babies whose mothers received chemotherapy during pregnancy.

Regarding the analysis of the circulating levels of angiogenic factors in pregnant women with BC, no previous data exist that evaluate these markers in pregnant women with breast cancer. Otherwise, a significantly higher incidence of small-for-gestational-age (SGA) is observed when chemotherapy is given during pregnancy, indicating a potentially toxic influence on placental development leading to placental malfunction (via incomplete trophoblast invasion into the uterus) [10,23]. This fact might explain why birthweight inversely correlates with the cycles of chemotherapy received during pregnancy in our PBC-CHT patients.

The results of our study suggest that the accumulative effect of chemotherapy treatments on placental function is reflected by an angiogenic antiangiogenic state at the end of pregnancy. PlGF and sFlt-1 levels correlate with the number of cycles and with birth weight. These findings suggest that placental insufficiency is the main trigger that accounts for adverse pregnancy complications observed in these mothers and babies. However, the sample volume is insufficient to allow a deep analysis of the number of cycles and the effect on the sFlt-1/PlGF ratio and sEng levels.

Before each chemotherapy cycle, clinicians may have to take into consideration maternal and fetal status. There are no clear guidelines for optimal pre-chemotherapy fetal and mother assessment. At our institution, we agreed to perform a full obstetric assessment including an abdominal study with Doppler velocimetry before each cycle on top of standard blood tests to rule out neutropenia that may prevent chemotherapy administration. If maternal status and fetal tests were fine, a new cycle of chemotherapy was administrated.

Given that all efforts should be made to delay delivery [24] until at least 35–37 weeks of gestational age to reduce risk of fetal morbidity associated with prematurity, our findings can be used as a useful clinical tool to better plan the timing of delivery in pregnant women affected by BC. sFlt-1:PlGF ratio will allow monitoring the placental function throughout the pregnancy in order to prevent obstetric complications. As values of the sFlt-1:PlGF ratio higher than 38 confer increased risk of obstetrical complications (PE/IUGR) and preterm labor, the medical team should increase the maternal and fetal well-being monitoring and follow-up of these patients [16]. Moreover, complementary, our study also suggests that normal levels of angiogenic factors in women exposed to chemotherapy during pregnancy may be useful tools to rule out an early complication and decide to proceed to administer an additional cycle of chemotherapy if deemed appropriate and contribute to lengthen some weeks of pregnancy. It is important to highlight that in our group of pregnant women with breast cancer, angiogenic factors were within normal values for gestational age until the end of pregnancy that increased compared to control pregnant women, although without reaching the values of pregnant women with preeclampsia.

According to European Society of Medical Oncology (ESMO) guidelines for breast cancer treatments diagnosed during pregnancy, the indications for systemic therapy should follow those for the non-pregnant setting, taking into consideration the gestational age at diagnosis and the expected date of delivery. Chemotherapy is generally safe beyond the first trimester of gestation; however, increased rates of premature delivery, growth retardation, and stillbirth have been reported and close monitoring of the pregnancy is recommended. Dose calculation should follow the standard procedures outside the pregnancy setting, acknowledging that the pharmacokinetics of some cytotoxic drugs might be altered during pregnancy, but no dose adjustments are recommended [11]. Our results do not allow a different approach. Hypothetical dose customization according to sFlt-1/PlGF ratio may reduce treatment efficacy. In our experience, we feel more comfortable delaying one cycle administration because of maternal or fetal needs than lowering chemotherapy dose.

Even some preclinical data have been reported on new formulations of anthracyclines (liposomal formulations) and taxanes (nab-paclitaxel) about the potential reduction in the crossing of the placental barrier in animal models [25], none of the guidelines or consensus for breast cancer treatment during pregnancy included these new formulations due to lack of data in humans. All patients in our study were treated with standard formulations, following current guidelines [11].

Potential limitations of this study include the small sample size (*n* = 12). However, taking into account the singularity of these patients, this set of data could be useful for hypothesis-generating and encourage further investigations on angiogenic and antiangiogenic factors as potential prognostic markers in breast cancer patients. Further studies may help to elucidate the correlation of angiogenic factors and the number of cycles or gestational age at starting.

## 5. Conclusions

Women with breast cancer treated during pregnancy showed an antiangiogenic state that correlates with the number of cycles administered. Our results suggest that angiogenic factors could be useful in the clinical obstetric management of these patients as they could anticipate placental complications that usually develop in these patients. More studies are needed to evaluate the potential causative effect of CHT on placenta function and development and its implication in adverse fetal and maternal outcomes observed in these patients. Because of the low incidence of BC during pregnancy, international prospective collaboration is needed for advancing the knowledge of optimal management and treatment.

## Figures and Tables

**Figure 1 cancers-13-00923-f001:**
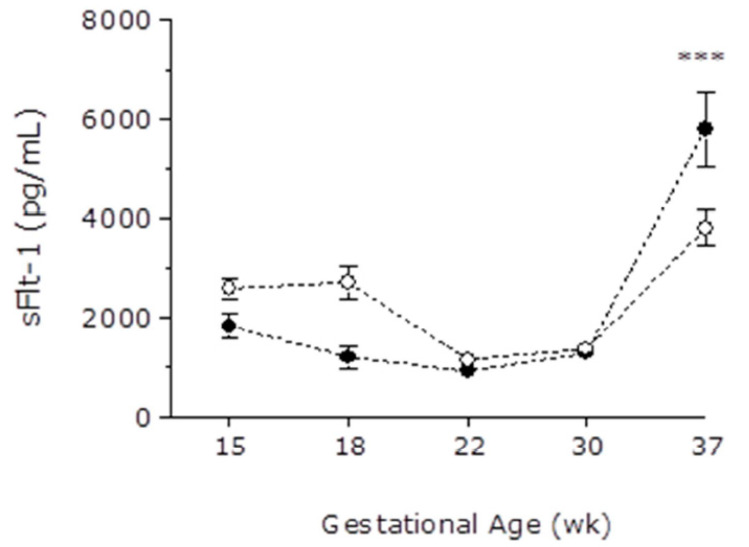
Maternal plasma concentrations of soluble fms-like tyrosine kinase (sFlt-1) in the control group (white points) and pregnant women affected by breast cancer (black points). Differences were analyzed by two-way analysis of variance. Data are presented as mean with standard error mean (SEM). ***, *p* < 0.001.

**Figure 2 cancers-13-00923-f002:**
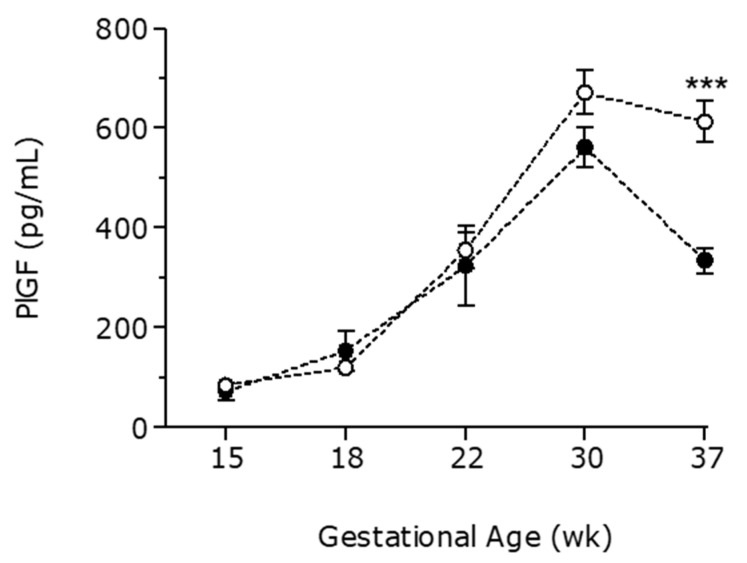
Maternal plasma concentrations of placental growth factor (PlGF) in the control group (white points) and pregnant women affected by breast cancer (black points). Differences were analyzed by two-way analysis of variance. Data are presented as mean with SEM. ***, *p* < 0.001.

**Figure 3 cancers-13-00923-f003:**
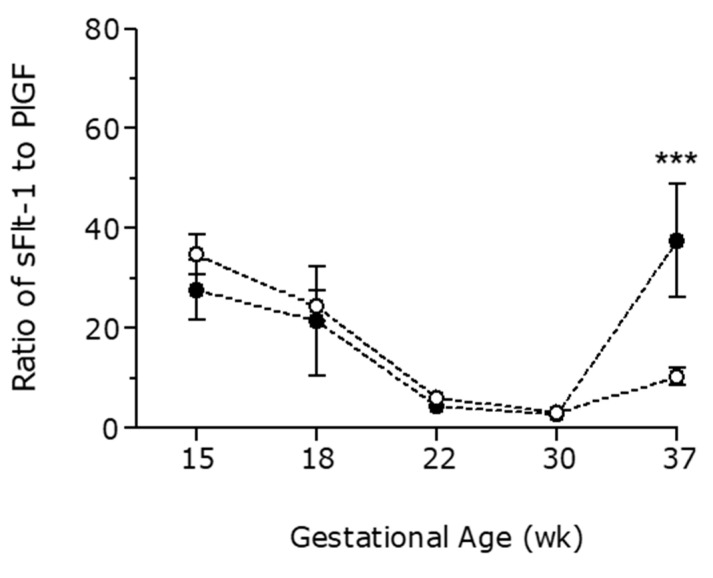
Ratio of sFlt-1 to PlGF in the control group (white points) and pregnant women affected by breast cancer (black points). Differences were analyzed by two-way analysis of variance. Data are presented as mean with SEM. ***, *p* < 0.001.

**Figure 4 cancers-13-00923-f004:**
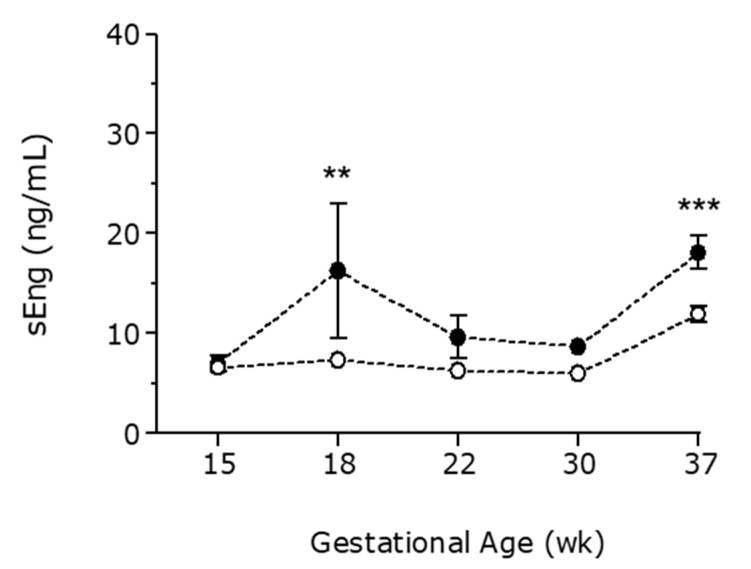
Maternal plasma concentrations of soluble endogline (sEng) in the control group (white points) and pregnant women affected by breast cancer (black points). Differences were analyzed by two-way analysis of variance. Data are presented as mean with SEM. **, *p* < 0.01; ***, *p* < 0.001.

**Figure 5 cancers-13-00923-f005:**
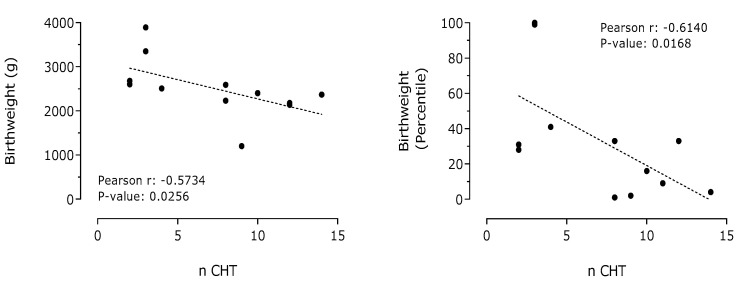
Correlation between cycles of chemotherapy administered (n CHT) and birth weight or percentile in pregnant women affected by breast cancer (*n* = 12).

**Figure 6 cancers-13-00923-f006:**
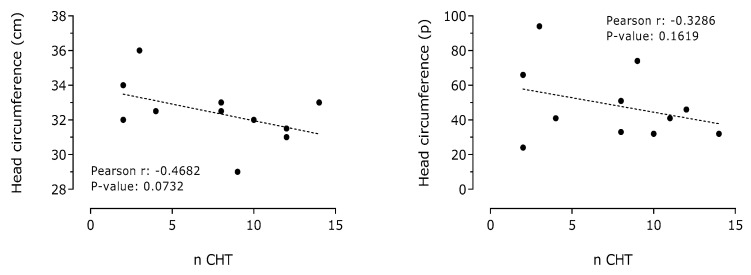
Correlation between number of chemotherapy cycles administered (n CHT) and neonate head circumference or percentile in pregnant women affected by breast cancer (*n* = 12).

**Table 1 cancers-13-00923-t001:** Demographic characteristics of study population.

	P-CTRL (*n* = 215)	PBC-CHT (*n* = 12)	*p*-Value
Maternal Age (years)	32.11 ± 5.12	37.08 ± 4.14	0.0012
BMI (Kg/m^2^)	24.27 ± 4.31	24.02 ± 4.28	0.7045
Caucasian	202 (64)	10(83)	0.1831
Conception			
Spontaneous	212 (99)	11 (92)	0.1965
Assisted	3 (1)	1 (8)	
Smoker	32 (15)	0 (0)	0.2248
Parity			
Nulliparous	114 (53)	6 (50)	1.000
Parous	101 (47)	6 (50)	

**Table 2 cancers-13-00923-t002:** Perinatal outcome of study population.

	P-CTRL (*n* = 215)	PBC-CHT (*n* = 12)	*p*-Value
GA at delivery (weeks)	38.9 ± 2.08	35.8 ± 2.00	<0.0001
Birth weight (g)	3232 ± 464	2512 ± 655	<0.0001
Birth weight (p)	54.1± 27.99	39.5 ± 29.61	0.0462
SGA	5 (2.3)	2 (16)	0.0465
Length (cm)	49.7 ± 1.75	45.9 ± 3.12	<0.0001
Length (p)	53.9 ± 27.4	39.4 ± 29.4	0.1148
Head circumference (cm)	34.4 ± 1.31	32.4 ± 1.76	<0.0001
Head circumference (p)	49.2 ± 30.4	48.5 ± 21.3	0.9602
1 min Apgar <7		2 (16)	
Days NICU		1(8)	
Days Hospital		2 (16)	

**Table 3 cancers-13-00923-t003:** Correlations between the number of cycles of chemotherapy administered, birth weight or percentile and angiogenic factors values at delivery.

	Number of Cycles		Birth Weight		Birth Percentile	
	**r**	***p* Value**	**r**	***p* Value**	**r**	***p* Value**
sFlt-1 (pg/mL)	0.6140	0.0222	−0.2498	0.2294	−0.3598	0.1385
PlGF (pg/mL)	0.1395	0.3412	0.2369	0.2416	0.1680	0.3108
Ratio sFlt-1 to PlGF	0.2471	0.2302	−0.2364	0.2427	−0.2273	0.2517
sEng (ng/mL)	0.5046	0.0567	−0.1691	0.3096	−0.3452	0.1492

## Data Availability

The data presented in this study are available on request from the corresponding author. The data are not publicly available due to specifications on the signed IC.

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
