# Peer review of "Evolution of Angiogenic Factors in Pregnant Patients with Breast Cancer Treated with Chemotherapy"

_cancers, 2021, doi:10.3390/cancers13040923_

Round 1

Reviewer 1 Report

The manuscript “Evolution of angiogenic factors in pregnant patients with breast cancer treated with chemotherapy” describes a cohort study to find a relation between sFlt-1 level and chemotherapy. The aim and the main finding of the study are well-reported, the knowledge provided, thanks to the data quality and the statical analysis, will certainly help to improve the pregnancy outcome of pregnant patients with breast cancer.

However, discussion and conclusion aren’t developed enough to valorize the good data. The following point could extend the discussion

-   the possibility to build customized chemotherapy for pregnant women, dose, Frequency,…

- the impact of the pharmaceutical formulation of Anthracycline and  Taxanes, as they are available with liposomal or nanoparticles formulation and some ex-vivo studies showed that liposomal based formulation has a different rate of transplacental passage.

- Collecting clinical data of sflt-1, is the cost justify the benefits...  

- the sFlt-1 level and preeclampsia management are studied in several recent papers, the gene delivery using specific drug delivery systems and blood dialysis are potential strategies for preeclampsia management        

For the conclusion, I understand that authors can’t build a larger conclusion because the sample is =12, but this is not a problem as it is justified by the limited availability on a regional level. So, the conclusion could mention the importance of international collaborations in collecting and sharing data regarding this particulate patient. Advanced data analysis methods could also be mentioned.  

Author Response

We, the authors, thanks reviewer 1 for their interesting remarks and comments that allow us to improve the quality of our paper. Here are our answers:

Point 1: the possibility to build customized chemotherapy for pregnant women, dose, Frequency,…

Answer to point 1: We thank the reviewer's comment. We have added a paragraph to highlight our point of view on a hypothetical dose customization (page 10, 4th paragraph, highlighted in yellow) and ref 25:

According to ESMO guidelines for breast cancer treatments diagnosed during pregnancy, the indications for systemic therapy should follow those for the non-pregnant setting, taking into consideration the gestational age at diagnosis and the expected date of delivery. Chemotherapy is generally safe beyond the first trimester of gestation however, increased rates of premature delivery, growth retardation and stillbirth have been reported and close monitoring of the pregnancy is recommended. Dose calculation should follow the standard procedures outside the pregnancy setting, acknowledging that the pharmacokinetics of some cytotoxic drugs might be altered during pregnancy, but no dose adjustments are recommended [25]. Our results do not allow a different approach. Hypothetical dose customization according to sFlt-1:PlGF ratio may reduce treatment efficacy. In our experience, we feel more comfortable delaying one cycle administration because of maternal or fetal needs than lowering chemotherapy dose.”

Point 2: the impact of the pharmaceutical formulation of Anthracycline and  Taxanes, as they are available with liposomal or nanoparticles formulation and some ex-vivo studies showed that liposomal based formulation has a different rate of transplacental passage

Answer to point 2: We thank the reviewer's comment. We have added a paragraph to highlight our point of view on a hypothetical dose customization (page 10, 5th paragraph, highlighted in yellow) and ref 26:

“Even some preclinical data has been reported on new formulations of anthracyclines (liposomal formulations) and taxanes (nab-paclitaxel) about the potential reduction in the crossing of the placental barrier in animal models [26], none of the guidelines or consensus for breast cancer treatment during pregnancy included these new formulations due to lack of data in humans. All patients in our study were treated with standard formulations, following current guidelines [25].”

Point 3: Collecting clinical data of sflt-1, is the cost justify the benefits..

Answer to point 3: We thank the Reviewer for his/her interesting remark; nowadays, sflt1/PlGF ratio is available in the majority of maternal-fetal high-risk clinics and tertiary hospitals in most developed countries and they constitute a biomarker for the management of preeclampsia in the routine pregnancy care and it has been proven that is cost-effective in women at risk for placental complications (https://www.nice.org.uk/guidance/dg23). Our data would help to include pregnant women receiving QT treatment into routine angiogenic factors screening and management. This observation has been already acknowledge in the conclusion section (Page 10, 2nd paragraph, highlighted in yellow):

sFlt-1:PlGF ratio will allow monitoring the placental function throughout the pregnancy in order to prevent obstetric complications. As values of the sFlt-1:PlGF ratio higher than 38 confer increased risk of obstetrical complications (PE/IUGR) and preterm labor, the medical team should increase the maternal and fetal well-being monitoring and follow-up of these patients

Point 4: the sFlt-1 level and preeclampsia management are studied in several recent papers, the gene delivery using specific drug delivery systems and blood dialysis are potential strategies for preeclampsia management

Answer to point 4: We completely agree with the Reviewer regarding possible therapeutic options for preeclampsia, however it is important to highlight that in our group of pregnant women with breast cancer, angiogenic factors were within normal values for gestational age until the end of pregnancy that increased compared to control pregnant women, although without reaching the values of pregnant women with preeclampsia. This information has now been included in the discussion section (page 10, end of 2nd paragraph, highlighted in yellow).

It is important to highlight that in our group of pregnant women with breast cancer, angiogenic factors were within normal values for gestational age until the end of pregnancy that increased compared to control pregnant women, although without reaching the values of pregnant women with preeclampsia

Point 5: For the conclusion, I understand that authors can’t build a larger conclusion because the sample is =12, but this is not a problem as it is justified by the limited availability on a regional level. So, the conclusion could mention the importance of international collaborations in collecting and sharing data regarding this particulate patient. Advanced data analysis methods could also be mentioned

Answer to point 5: We completely agree with the Reviewer, international sharing and collaborations are needed for better knowledge of this non frequent entity. We added a sentence at the end of the conclusion paragraph (Page 10, final paragraph, highlighted in yellow).

Because of the low incidence of BC during pregnancy, international prospective collaboration is needed for advancing in the knowledge of the optimal management and treatment

Reviewer 2 Report

It’s very interesting study and I agree with Authors that it is first communication about the influence of chemotherapy during pregnancy on angio/antiangiogenic status of placenta. Ratio sFlt-1 to PlGF rather than sFlt-1, PlGF and sEng alone, is known in clinical practice as a good marker of placental functioning and in consequence of preeclampsia.

The idea is very interesting but some questions must be explained before publication:

  1. the important problem is not only number of cycles of CHT but also the moment of beginning CHT, in my opinion the analysis of this question inside the study group is needed

  1. the correlation n CHT cycles not only with some clinical conditions but also with sFlt-1 to PlGF ratio would be interesting and give better opportunity for conclusion about the mechanisms of influence of CHT on the risks of pregnancy complications

  1. the Authors prepared the large control group but didn’t used it for more deep analysis. In the manuscript there are only the comparison the clinical features but not the markers of angio/antiangiogenetic status between the groups. It is the separate problem if the control group – significant younger than the study group could be useful for correct analysis.

  1. The conclusion about the potential role of the angiogenic factor in management of pregnant women with CHT is interesting but completely speculative.

Author Response

We, the authors, thanks reviewer 2 for their interesting remarks and comments that allow us to improve the quality of our paper. Here are our answers:

Point 1: “the important problem is not only number of cycles of CHT but also the moment of beginning CHT, in my opinion the analysis of this question inside the study group is needed

Answer to point 1: We thank the reviewer's comment. We acknowledge that moment of chemotherapy administration may influence pharmacokinetic of the compounds and may alter the sFlt-1 or PIGF levels, but due to the limited number of samples available for the analysis related with the low frequency of breast cancer diagnosed during pregnancy, we analyzed all samples available and report number of chemotherapy cycles for each patient (that depends on the moment of breast cancer diagnosis). In this study, only 2 patients had been diagnosed during the third trimester. We have analyzed the correlation between the number of cycles and sFlt-1/PlGF ratio and sEng levels. In our results, there is a broad dispersion provably because small sample volume. By contrast, there is the clearest correlation with gestational weeks as we show in our article. The higher is gestational age, the higher are the placental needs and the higher are the differences with controls. That is why we focused our attention on gestational age instead of the number of cycles. We added a sentence in the discusion (Page 9, 4th paragraph, highlighted in green).

However, the sample volume is insufficient to allow a deep analysis of the number of cycles and the effect on the sFlt-1/PlGF ratio and sEng levels.

Point 2: “the correlation n CHT cycles not only with some clinical conditions but also with sFlt-1 to PlGF ratio would be interesting and give better opportunity for conclusion about the mechanisms of influence of CHT on the risks of pregnancy complications

Answer to point 2: We thank the reviewer's comment. As we discussed in Answer to point 1 the sample volume is inadequate for this analysis.  Although there is a positive correlation (see images for the reviewer), we have been focused on the gestational age because is the clearest interpretation. Future studies may help to show the correlation between the number of cycles and sFlt-1/PlGF ratio and sEng levels. We added a sentence at the end of the discussion (Page 10, 4th paragraph, highlighted in green)

Point 3: “the Authors prepared the large control group but didn’t used it for more deep analysis. In the manuscript there are only the comparison the clinical features but not the markers of angio/antiangiogenetic status between the groups. It is the separate problem if the control group – significant younger than the study group could be useful for correct analysis

Answer to point 3: The comparison of the markers of the angiogenic status between groups is presented in the figures of the manuscript (figure 1, 2, 3 and 4). In these figures we show the plasmatic concentration of angiogenic factors throughout pregnancy and the antiangiogenic status detected in the study group in the third trimester.

The reviewer points out that the control group is significantly younger than the study group, which is right just looking at the P-value, although the reported means and standard deviations, suggests otherwise (Table 1). The test is reporting strong significance because of the big difference in the sample sizes between the two groups; therefore, future studies with equal sample sizes would be more accurate at this point

Point 4: “The conclusion about the potential role of the angiogenic factor in management of pregnant women with CHT is interesting but completely speculative”

Answer to point 4: We present interesting results with a short sample of patients with breast cancer during pregnancy. Regarding the observed differences and taking into account the complexity of management, we conclude that sFlt-1/PlGF ratio and sEng levels may help. It is impossible with a low incidence of BC during pregnancy to achieve enough samples in a regional model. We suggest that an international prospective study is needed to confirm the clinical utility of this approach. This hopefully preliminary result justifies this international prospective study. We added a sentence at the end of the Conclusion paragraph (Page 10, 6th paragraph, highlighted in yellow)
